# Accurate and fast prediction of radioactive pollution by Kriging coupled with Auto-Associative Models

Raphaël Périllat[1], Sylvain Girard[1], and Irène Korsakissok[2]

[1]Phimeca, Paris, France
[2]ASNR, Paris, France

**Correspondence:** Raphaël Périllat (perillat@phimeca.com)

**Abstract.** Uncertainty estimation is a key issue in nuclear crisis situations. Probabilistic methods for taking uncertainties into account in assessments are often costly in terms of the number of simulations and computation time. This is why emulation methods, which enable rapid estimation of numerical model outputs, represent a promising solution. However, in the context of radioactive dispersion modeling, existing emulators are mostly limited to scalar outputs. In a crisis context, decisions are often based on dose maps, which are mathematically represented by high-dimensional data. In this study, we use the Auto-Associative Model method to reduce the dimension of dose results, and then predict these reduced representations using Kriging. We also compare this prediction method with others used by the French Nuclear Safety and Radiation Protection Authority (ASNR) to predict the consequences of a nuclear accident.

## 1 Introduction

### 1.1 Context

In the event of a nuclear accident, numerical simulations of atmospheric dispersion are used to predict the territories potentially impacted by radioactive releases. The French Authority for Nuclear Safety and Radiation Protection (ASNR) develops and uses atmospheric dispersion models embedded within its operational crisis platform called C3X to perform these calculations (Tombette et al., 2014). These simulations are used to infer operational indicators such as the maximum distance from the source where a dose threshold will be exceeded. The thresholds may be, for instance, regulatory protective action guide levels that could trigger protective actions such as population evacuation, sheltering, stable iodine prophylaxis or food restrictions. Although the dose guide levels differ from one country to another, this approach based on zones of threshold exceedance is widely used, as described in IAEA's publications on emergency preparedness and response (int, 2021).

Such evaluations are subject to uncertainties due to lack of information on the installation's status, meteorological forecast uncertainties, and models' approximations (Leadbetter, S.J. et al., 2020; LE et al., 2021). Prediction errors can induce two kinds of wrong decisions: either insufficient population protection zones, where a threshold exceedance occurs but was not predicted, or unnecessary actions zones where a threshold exceedance is forecast but does not come true. While the detriment to the population in the former case is obvious, leading to the use of conservative evaluations designed to avoid this situation at all costs, limiting evacuation and other restrictions where possible is also desirable, as these actions may have a high and

potentially long-term economical and health cost (Nomura et al., 2013, 2016). A better quantification of uncertainties may help refine the hypotheses and potentially reduce the margins of the conservative assumptions, while ensuring a sufficient population protection. In the approach applied by the ASNR's emergency center, the very first response generally relies on pre-calculated scenarios, whose data are gathered in an "Accident Type Sheet" (ATS). This database relies on calculations carried out in the preparedness phase for a number of accidental scenarios and for selected weather situations described by a few parameters (wind direction and speed, atmospheric stability, rain), assumed to be constant in time and homogeneous over the simulation domain. In a second step, ASNR uses its local scale Gaussian puff atmospheric dispersion model called pX (Korsakissok et al., 2013; El-Ouartassy et al., 2022) to obtain predictions that correspond more closely to the actual accidental and meteorological situation.

Forecasting tools must be compatible with emergency response time constraints, when the first evaluations should be provided typically within one hour after the alert. This timing includes not only the computational time required to set up and run the simulations themselves, but also the time required to gather meteorological forecasts and source term assessment, analyse the results and communicate them to decision makers. Thus, a numerical model such as pX requiring typically a few minutes to run, can be used for a single, deterministic estimation. However, the computation time required to account for uncertainties by using hundreds of simulations does not fit with these operational constraints.

## 1.2 Emulation and dimension reduction

An emulator is a surrogate model that approximates the output of a computationally intensive physical model while being much faster to evaluate. Emulators are often constructed by interpolating scalar outputs from a set of precomputed simulations.

In radiological emergency contexts, emulators enable several practical applications:

– Enlarge the pre-calculated scenario database by including a variety of input parameters, allowing to evaluate results for a larger range of meteorological situations than those considered in the ATS;

– Replace the original model in the case of uncertainty estimation (Le et al., 2018) or sensitivity analyses (Girard et al., 2016), where several hundreds of simulations are needed to perturb the model inputs in order to obtain a large number of outputs;

– Thanks to their speed, they allow interactive exploration of the input space, with a graphical interface where it is possible to vary the model inputs to observe their influence on the output. Such a tool can be used for education and training purposes, in order to demonstrate the influence of input parameters on the outputs and to help making "reasonably conservative" evaluations of uncertain parameter values.

However, the output of dispersion models is typically a spatial map—high-dimensional and structured—whereas most emulators are designed for scalar outputs. One naive solution would be to build an emulator for each grid point, but this approach ignores spatial correlations and is computationally inefficient.

In the field of atmospheric dispersion modelling, emulators have been used to predict spatio-temporal average quantities, values at a monitoring point (Le et al., 2018; Girard et al., 2016), or maximum exceedance distances (Périllat et al., 2020). Fitting an emulator for each grid point of a two-dimensional map would be both difficult to compute and to implement, as it would ignore the spatial correlations inherent in the data. For these reasons, emulating a spatial map requires a first step of dimension reduction.

The most widely used dimension reduction method is Principal Component Analysis (PCA) (Jolliffe and Cadima, 2016; Jolliffe, 2002). It consists in projecting a set of points onto a vector subspace in a least squares optimal way, in order to obtain the most faithful representation of this set of points in a reduced dimensional subspace. It has been applied in the specific domain of atmospheric dispersion, sometimes in combination with emulation (Burgin et al., 2017; Le et al., 2018; Mallet et al., 2018; Swallow et al., 2017; Lumet et al., 2025). However, being a linear approximation, PCA fails to encode sets of maps that are too different from one another.

The Auto-Associative Model (AAM) is an extension of PCA that captures nonlinear relationships in the dataset (Girard, 2000). Instead of projecting the data onto a linear subspace, AAM approximates the dataset by a low-dimensional differentiable manifold. It does so through two complementary mappings: a projection function that reduces the dimensionality, and a recovery function that reconstructs the original data from the reduced coordinates. These mappings are typically estimated using spline regression, allowing the model to learn smooth nonlinear variations in the dataset.

AAM can be seen as a constrained form of autoencoder, where both the architecture and the optimization strategy are adapted to small sample sizes and structured data, such as spatial fields. Compared to PCA, which assumes linear combinations of orthogonal basis vectors, AAM provides better reconstruction when data exhibit spatial transformations like shifts, deformations or plume rotations — situations that frequently occur in dose maps produced by atmospheric dispersion models.

AAM has already been applied once in this context, to the dimension reduction of radiological dispersion maps (Girard et al., 2020). In that study, it was shown that AAM could recover over 78% of the variance using only 2 nonlinear coordinates, while PCA required 6 components to achieve the same reconstruction quality. The difference was particularly notable for simulations with varying wind directions, where PCA struggled to encode rotated structures. Visual comparisons confirmed that AAM-preserved spatial features more faithfully.

The present paper presents the first combination of AAM with emulation, applied to the prediction of dose maps in the event of an accidental release of radioactive materials into the atmosphere. We introduce the case study in Section 2, the methodology in Section 3, and the results in Section 4. We develop and validate the AAM independently in Section 4.1.1, the Kriging model in Section 4.1.2, and finally the complete emulation framework in Section 4.1.3. Validation is performed on an operational scenario for predicting threshold exceedance areas. The performance of the proposed approach is compared to alternative prediction methods in Section 4.2. Finally, in Section 5, we illustrate the practical applications enabled by the emulator, particularly in terms of real-time response, probabilistic risk assessment, and decision support.

## 2 Case study

We simulated the result of a primary breach leading to a total core meltdown in one hour of a 1300 MWe Pressurized Water Reactor. This accidental scenario is one of the pre-calculated scenarios leading to the exceeding of protective action guide levels over significant distances. We used the pX Gaussian puff dispersion model with the Doury diffusion model in neutral atmospheric stability and with a meandering wind coefficient of 3. The meandering wind coefficient is a multiplicative factor applied to the diffusion model and designed to account for wind direction variations that occur during the time span of the release and are not taken into account by the meteorological inputs.

We focused here on 2D maps of thyroid inhalation equivalent dose, 24 hours after the beginning of the releases. In France, stable iodine prophylaxis is related to a dose criteria of 50 mSv to the thyroid. We used a polar mesh with smaller cells close the source, where there are strong spatial dose variations. The nodes of the mesh are distributed on 36 angles between 0 and 360° and on 61 different radii from 500 m, increasingly spaced from each other as we move away from the source, until a distance of 30 km.

Thus, the output data has a dimension of 2196. The case study is stationary: inputs are assumed to be constant in time and space. This is not generally the case for meteorological variables, but is consistent with the simple situations on which pre-calculated sheets are based. We considered six sources of uncertainty as inputs of the model, which are listed in Table 1. Two of them are related to the meteorological situation: the wind module and rainfall rate; two uncertain parameters, the source amplitude (a multiplicative factor applied to the source term computed for the chosen accidental scenario) and release height, describe the source term characteristics; and the two last parameters are used to define radionuclide deposition rates, for iodine and others.

The choice of uncertain parameters and their ranges of variation is not representative of the full range of possible situations. They were chosen to be representative of the usual range of values encountered during emergency exercises, to build and validate a proof of concept that can then be extended to other situations and scenarios. For instance, the source term computed for the accidental scenario comprises 224 radionuclides, each of them being associated with a release rate as a function of time. It was computed using conservative assumptions regarding the quantity of radioactive materials emitted into the atmosphere. Therefore, the multiplicative factor applied to these quantities is assumed to vary between 10 and 100%, as the evaluation of the installation status at the time of the accident is more likely to lead to a downward revision of the source term.

The ranges of variation for each input parameter were selected based on operational knowledge and literature values, as well as their relevance in emergency scenarios:

- **Wind module**: Varied between 0 and 10 m/s. This range covers typical near-surface wind speeds encountered in most meteorological conditions relevant for nuclear dispersion scenarios. Wind speeds above 10 m/s are less frequent and usually associated with lower concentrations, therefore less likely to trigger threshold exceedance.

- **Rain intensity**: Varied between 0 and 10 mm/h, covering the range from no precipitation to heavy rain. This interval includes typical rain intensities relevant for wet deposition modeling in nuclear dispersion scenarios. Rain intensities above 10 mm/h are less frequent and often short-lived, and are therefore not prioritized in standard emergency simulations.

– **Release height**: Varies from 0 to 100 m. A value of 0 m corresponds to a ground-level release, which is relevant for some accident scenarios such as leaks near the base of a reactor building. The upper bound of 100 m corresponds approximately to the height of the ventilation stacks or highest building points in most nuclear facilities, and thus reflects realistic release heights for stack emissions or elevated plumes. Higher plume emissions would not lead to any threshold exceedance at ground level for the accidental scenario considered.

– **Source term amplitude**: From 10 to 100 %, reflecting the uncertainty in estimating the actual quantity of radioactive material released. It is common practice during an emergency to revise the source term amplitude downward as more information becomes available about the plant's condition and containment efficiency.

– **Deposition velocities**: For iodine, a range of $1 \times 10^{-5}$ to $1 \times 10^{-2}$ m/s is used, and for other elements, from $5 \times 10^{-4}$ to $5 \times 10^{-3}$ m/s. The range of variation was derived from a literature review (Baklanov and Sørensen, 2001) and consistent with previous studies (Girard et al., 2014).

**Table 1.** Input parameters and ranges of variation for the construction of emulators.

| 70 Input variable | Range of variation | Units |
|---|:---:|:---:|
| Wind module | [0, 10] | $\text{m.s}^{-1}$ |
| Rain intensity | [0, 10] | $\text{mm.h}^{-1}$ |
| Release height | [0, 100] | m |
| Source term amplitude | [10, 100] | % |
| Deposition velocity of iodine | $[1 \times 10^{-5}, 1 \times 10^{-2}]$ | $\text{m.s}^{-1}$ |
| Deposition velocity of other elements | $[5 \times 10^{-4}, 5 \times 10^{-3}]$ | $\text{m.s}^{-1}$ |

## 3 Emulation method

### 3.1 Auto-Associative Models

"Reducing the dimension" of an ensemble embedded in a high dimensional vector space consists in building an associated ensemble, with a lower dimensional coordinate system. A rough definition of topological dimension would be "the minimum number of variables needed to represent a set" (Fukunaga and Olsen, 1971). More rigorously, we must choose the nature of the associated sets in order to have a precise definition of "coordinate system", for example the one recalled by (Milnor, 1997) for differentiable manifold.

Given a set $G \subset \mathbb{R}^m$, with a large $m$, we try to construct the approximate set $A \subset \mathbb{R}^m$ in bijection with the vector space $C \subset \mathbb{R}^l$, with $l$ small:

$$G \subset \mathbb{R}^m \xrightarrow{\psi} C \subset \mathbb{R}^l \overset{\chi^{-1}}{\underset{\chi}{\leftrightarrows}} A \subset \mathbb{R}^m \tag{1}$$

The Auto-Associative Models (Girard and Iovleff, 2008) is a nonlinear method of dimension reduction. The term "linear" here means that the approximating space $A$ would be a sub-vector space and $\chi \circ \psi$ an orthogonal projection. In contrast, in the "nonlinear" case, the approximating space $A$ is a differentiable manifold and $\chi \circ \psi$ a composition of orthogonal projections by a nonlinear $\chi$ function.

## 3.2 Kriging

Kriging is a spatial interpolation method and the core of geostatistics. It was originally designed for optimizing gold mining (Chilès and Delfiner, 1999) by inferring from a few boreholes the spatial distribution of gold grades over the whole mining field. Kriging becomes an emulation method by replacing the spatial coordinates by the model inputs, and gold grade by the scalar model output.

The Kriging emulator predicts the output at a new input point $x$ as a linear combination of $N$ known outputs $f(x_i)_{i=1}^N$ at previously simulated input points $x_i$. The weights $w_j$ of this linear combination are chosen to minimize the variance of the prediction error, under the assumption that the output is a realization of a Gaussian process. This Gaussian process is assumed to be second-order stationary, meaning its mean is constant and its covariance between two input points depends only on their relative position.

The weights are the solution to the following system:

$$\forall i \in \{1,\ldots,N\}, \quad \sum_{j=1}^N w_j K(x_i,x_j) + \lambda = K(x_i,x), \tag{2}$$

$$\sum_{j=1}^N w_j = 1. \tag{3}$$

where $K(\cdot,\cdot)$ is a positive-definite covariance kernel, $\lambda$ is a Lagrange multiplier enforcing unbiasedness, and $x$ is the target point. The predicted centered value is then $\hat{f}(x) = \sum_{j=1}^N w_j f(x_j)$. The mean of the process can also be estimated by a generalized least squares regression.

In this work, we use the `DiceKriging` R package (Roustant et al., 2012), which implements Kriging metamodels for deterministic computer codes. We use a product of stationary covariance kernels, one for each input variable, and estimate the correlation lengths via maximum likelihood optimization.

A comprehensive introduction to Kriging for emulation can be found in the textbook *Gaussian Processes for Machine Learning* (Rasmussen and Williams, 2006), and an application to atmospheric dispersion modeling is described in (Girard et al., 2016).

We used the R package `DiceKriging` (Roustant et al., 2012), which provides a choice of second order stationary co-variance kernels. We used a tensor product of identical kernels, one for each input variable. We fitted the parameters of these kernels (a characteristic length of spatial dependency for each input variable) by maximum likelihood estimation.

### 3.3 Putting it into practice

We performed 2548 simulations uniformly sampling the 5 dimensional input space of the release height, the wind module, the rain intensity and the two deposition velocity (their range of variation is given in table 1). The output dose being linear with the amplitude of the source term, we completed the simulation sample using two random amplitudes for each input vector, thus covering the whole 6 dimension input space with a total of 5096 points. 4096 of them are used to train the AAM, while the other 1000 were used to fit the model.

Given that the dose values vary by up to a factor of $10^5$ depending on distance, we applied the method to their logarithm to better reveal relative variations. The AAM reduced the dimension of the results to nine coordinates, whereas the initial data dimension was 2196, corresponding to a grid mesh of size $36 \times 61$. Using fewer than nine coordinates resulted in larger errors, while higher-dimensional approximations increased computation times with only moderate improvements in results. Since the final goal of this parameterization is to focus on the zones where a dose threshold $th$ is exceeded, we also truncated the logarithm of the doses: any values below $\log(th)/2$ were set to $\log(th)/2$. Thus, small dose variations do not disturb the parameterization by AAM.

Kriging is then used to create emulators. The emulator combined with dimension reduction can then be used to predict an output map for any new input vector (see Figure 2). The AAM can finally associate to these 9 scalars a two-dimensional map of inhalation dose. Each of the 9 AAM coordinates is predicted independently using a separate Kriging model, allowing us to capture the specific response behavior associated with each latent component.

The number of AAM coordinates was chosen based on a trade-off between reconstruction accuracy and model complexity. We performed a systematic validation by varying the number of coordinates from 1 to 12 and computing the relative recon-struction error on a separate test set. As shown in Figure 1, the relative error decreases sharply with the number of coordinates up to 8–9, beyond which additional coordinates yield only marginal improvements. The interquartile and 5–95% ranges also stabilize at that point, indicating that the reconstruction quality no longer significantly benefits from added complexity. Based on this analysis, we retained 9 coordinates as a good compromise.

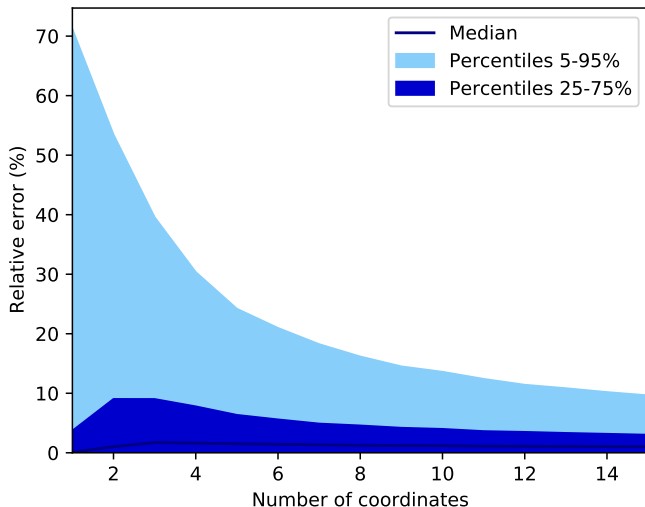

**Figure 1.** Evolution of the relative reconstruction error (in %) as a function of the number of AAM coordinates. The solid line represents the median error on the test set; the shaded areas correspond to the 25–75% and 5–95% percentiles.

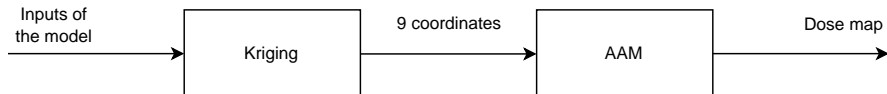

**Figure 2.** Dose map prediction process using emulation coupled with AAM.

## 4 Results

### 4.1 Validation

An additional simulation sample was built to test the reliability of the emulator. We drew $M = 1000$ random new points in our 6-dimensional input space and then run the computational model $M$ times with input parameters corresponding to these draws. We obtained $M$ output results that were compared with the maps reconstructed after dimension reduction with AAM (section 4.1.1) and with emulators predictions (section 4.1.2), then with the combination of the two methods (section 4.1.3).

### 4.1.1 Validation of the dimension reduction by AAM

We assessed the validity of the dimension reduction step by comparing the maps two by two for our test sample, as in figure 3, and by calculating the Figure of Merit in Space (FMS) of the dose criteria exceedance isolines for each of these maps. This score is calculated by dividing the area of the intersection of the two surfaces A and B by the area of the union of the two:

$$\text{FMS} = \frac{A \cap B}{A \cup B}$$

The FMS of two very similar surfaces approaches 1, while when two isolines have little surface in common it approaches 0. This metric has been used in previous studies of radioactive dispersion, such as the European CONFIDENCE project (Bedwell, P. et al., 2020), where it helped quantify the spatial agreement between predicted and reference dose areas.

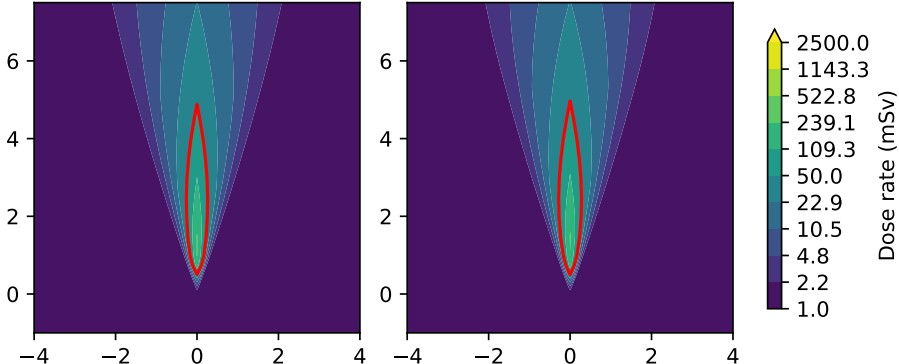

**Figure 3.** Inhalation dose map and 50 mSv guide-level exceedance isolines for a model output (left) and for the approximation of this output by AAM (right).

Figure 4 presents a histogram of the FMS values, showing that the areas enclosed by the isolines exceeding 50 mSv are well preserved by the AAM approximation. In 75.5% of the cases of the testing samples, the FMS is greater than 0.8, indicating a strong agreement between the AAM prediction and the reference model. In 11.7% of the cases, the FMS could not be computed because no threshold exceedance was observed, meaning that no isoline was formed. Overall, these results confirm that the predicted isolines are very close to those produced by the original model.

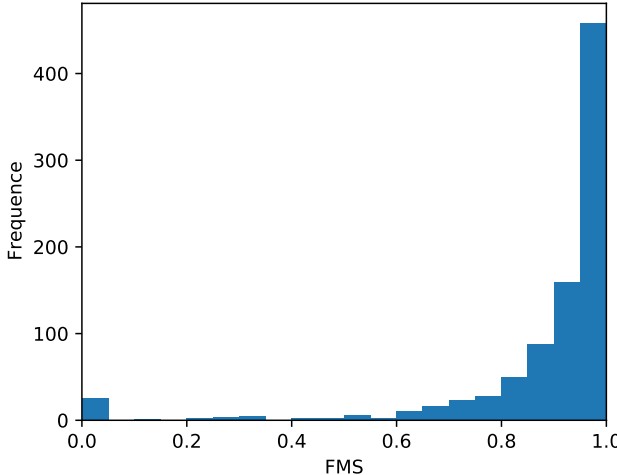

**Figure 4.** Histogram of the FMS that compares the threshold exceedance zones for the model outputs with AAM approximations across the testing samples. 75.5 % of the FMS are between 0.8 and 1, which implies that the projected results are very often similar to those obtained by simulation.

### 4.1.2 Validation of the interpolation by Kriging

Figure 5 shows how well each scalar is predicted. The closer the set of points is to the line $y = x$, the better the prediction.

We quantified the prediction error with the Standardized Mean Squared Error (SMSE). For a set of $N$ observed points $(x_i)_{i \in [1,N]}$ and a set $(\hat{x}_i)_{i \in [1,N]}$ of estimated points, the SMSE is defined as follows:

$$\text{SMSE} = \frac{\sum_{i=1}^{N} (x_i - \hat{x}_i)^2}{\sum_{i=1}^{N} (x_i - \bar{x})^2},$$

where $\bar{x}$ is the mean of $(x_i)_{i \in [1,N]}$.

**Table 2.** SMSE and 95% quantile of relative absolute error for each score

| Score | SMSE | Q95 rel. abs. error |
|-------|------|---------------------|
| 01 | 9.939e-04 | 1.219e-01 |
| 02 | 6.482e-03 | 9.413e-01 |
| 03 | 1.466e-02 | 1.811e+00 |
| 04 | 7.065e-02 | 4.368e+00 |
| 05 | 1.228e-01 | 4.044e+00 |
| 06 | 6.076e-02 | 2.078e+00 |
| 07 | 1.604e-01 | 3.231e+00 |
| 08 | 5.114e-01 | 2.268e+00 |
| 09 | 1.001e+00 | 3.134e+00 |

We can observe in Table 2 that the first three coordinates are reconstructed with excellent accuracy (very low SMSE). The SMSE increases progressively with the score index, reaching high values for the last two scores. This degradation is largely due to one outlier point in the test sample, which heavily impact the SMSE because it is based on squared errors.

To provide a more robust evaluation of the prediction quality, we also computed the 95% quantile of the relative absolute error. This metric gives an upper bound on the error affecting most points, without being overly sensitive to rare extreme values. As shown in Table 2, the Q95 scores remain acceptable even for the least accurate coordinates, confirming that the predictions remain generally reasonable despite isolated errors. These last coordinates bring fine details to the reconstruction and were therefore retained to preserve the quality of the final maps.

### 4.1.3 Validation of the emulator that combines the two methods

With the AAM, this prediction of the 9 scalars can be transformed into a 2D dose map, which can be used to determine a threshold exceedance isoline. The succession of the two methods thus allows to convert the model inputs into a decision aiding map, defined by an isoline, to estimate whether or not a guide-level might be exceeded.

Comparisons between simulations and associated emulator predictions can be classified into four cases:

- Case 1: Dose maps that are well reconstructed by the method, as shown in figure 6 (a) and (b). The shape of the isoline is preserved, and its size is similar to that given by the simulation, resulting in a well-estimated maximum distance.

- Case 2: Dose maps that are well reconstructed, but the isolines can differ slightly in scale from what is expected, as illustrated in figure 6 (c) and (d).

- Case 3: Dose maps that are less well reconstructed, where the general shape of the isoline is not accurately reproduced. This typically occurs under low wind conditions, as shown in figure 6 (e) and (f).

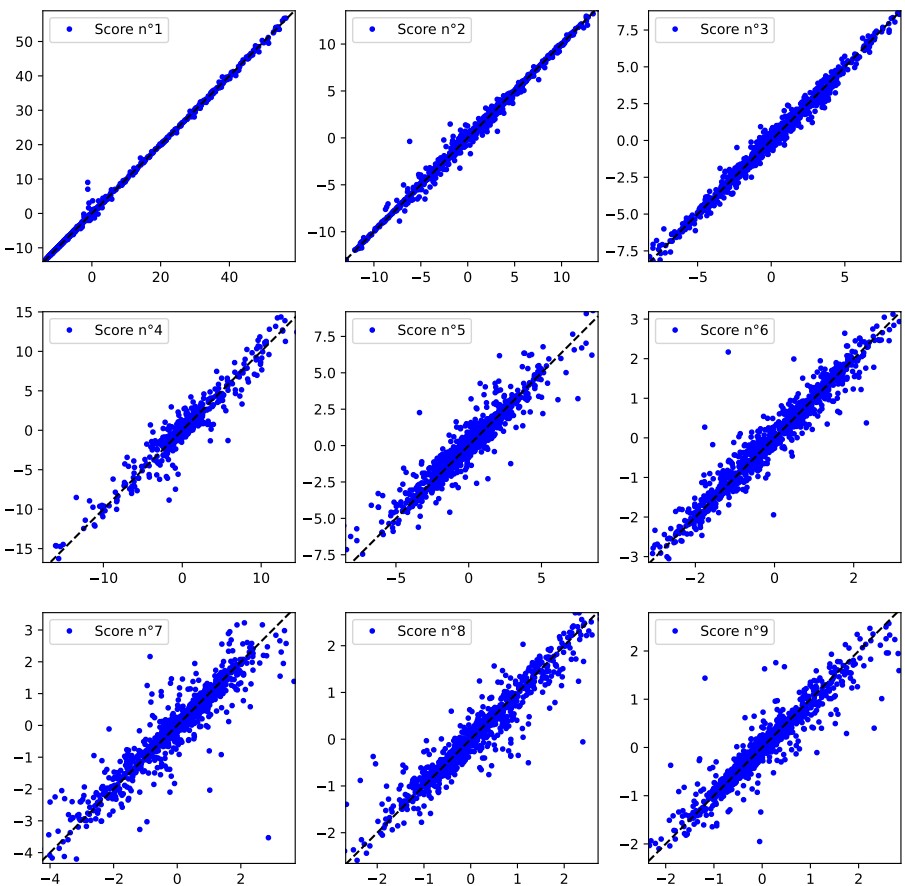

**Figure 5.** Comparison of the prediction of the emulator (in y-axis) with the target obtained by simulation (in x-axis). Each graph represent one coordinate of the AAM.

– Case 4: Dose maps that approach the threshold value with little or no exceedance, but the predicted and simulated isolines may differ significantly. Figure 6 (g) and (h) illustrates this: although the two dose maps appear similar, a slight difference in intensity can cause a noticeable discrepancy in the exceedance isoline prediction.

We evaluated the adequacy of the simulated and predicted surfaces by calculating the FMS of our test sample simulations. Figure 7 represents the histograms of the FMS. We can notice a slight degradation compared to Figure 4. However 70.9 % of the FMS are between 0.8 and 1, which implies that the predicted isolines are very often similar to those obtained by simulation (Case n°1 and n°2 described earlier). Also, 11.5 % of the FMS cannot be calculated, because both of the simulation and emulator detect no threshold exceedance. Those cases correspond also to a good prediction of the emulator. We note that 2.8 % of the FMS are equal to zero, which correspond to the problem of the non-reached threshold mentioned previously (Case n°4). The last 14.8 % intermediate FMS correspond to badly reconstructed isolines (Case n°3), for instance when the wind module is low, which does not necessarily mean that the surface error is large, because the FMS is a relative score. Excluding

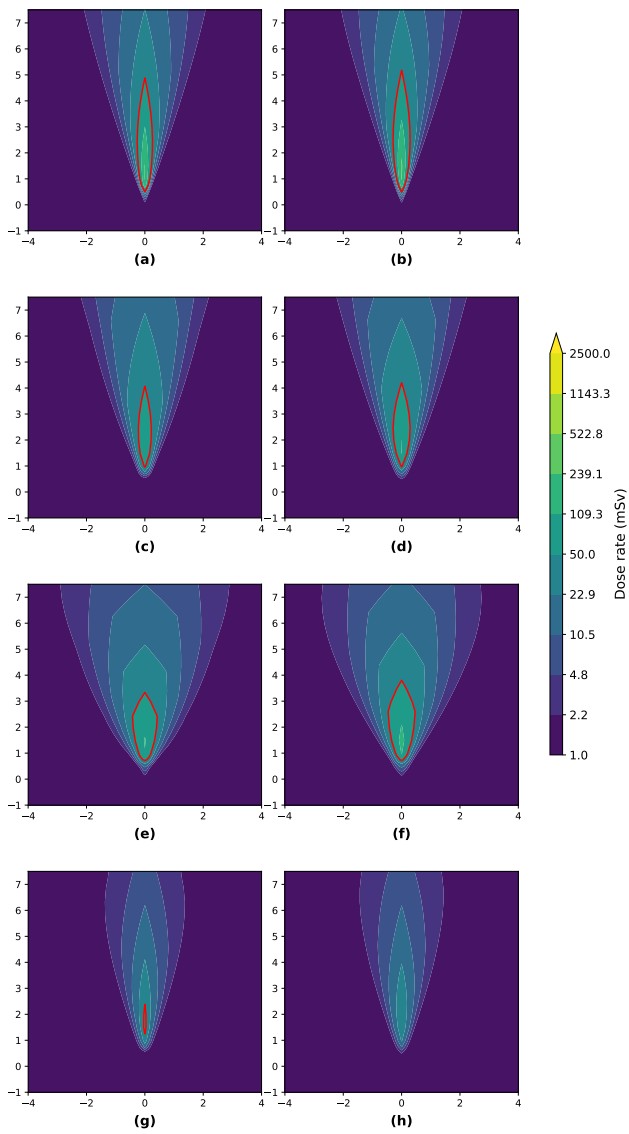

**Figure 6.** Examples of inhalation dose maps for simulated results by the original physical model (left) and emulator predicted results (right).

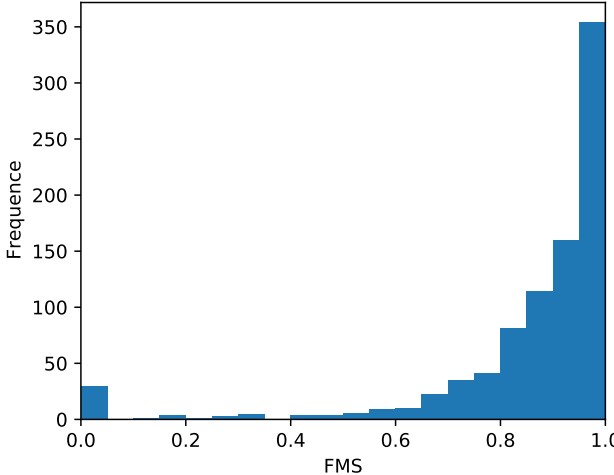

**Figure 7.** Histogram of the FMS that compares the threshold exceedance zones for the model outputs with the emulator outputs across the testing samples. 70.9 % of the FMS are between 0.8 and 1, which implies that the predicted results are very often similar to those obtained by simulation.

the cases where no threshold is exceeded, badly reconstructed areas amount to 17.6 %, which means that more than 80 % of the threshold exceedance areas are correctly forecast.

## 4.2 Comparison to other prediction methods

We benchmarked our new prediction method against two state of the art procedures:

- At the start of a crisis, a first estimate is derived from the ATS, yielding orders of magnitude and first results for distance
and angular aperture of exceedance. Then simulations are performed with the pX model in order to predict a guide-level exceedance zone and estimate a maximum distance of threshold exceedance. An angular aperture is also estimated from pre-computed tables depending on wind, atmospheric stability, and meandering wind factor. This simulated maximum distance, associated with this angular aperture obtained without emulation, allows to deduce a portion of circle which corresponds to the zone for which decisions are recommended.

- In a previous study (Périllat et al., 2020), an emulator was created to directly predict, without using AAM, the maximum distance of the threshold exceedance given by the model, as well as the angular aperture of this zone. Kriging was used to estimate those two geometrical parameters from the original model pX.

These two methods will be referred to as the 'ATS estimator' and the 'Emulator of geometrical parameters' in the remainder of this paper.

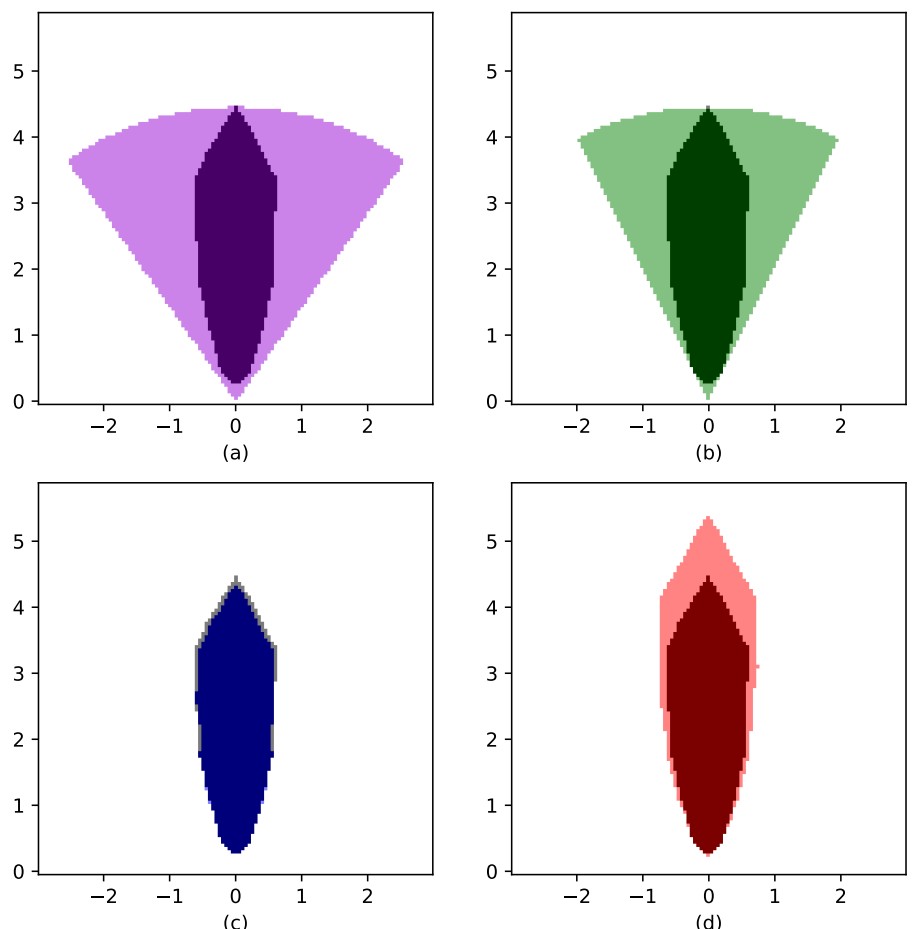

**Figure 8.** Guide-level exceedance zone, for one simulation of the test sample, obtained by the different estimation methods: the ATS estimator (a), the emulator of geometrical parameters (b), prediction by our emulator which coupled Kriging to AAM (c), prediction by the same emulator with a lower threshold (d). The zone of guide-level exceedance given by the Gaussian puff model pX is superimposed to each graph (darker color).

Figure 8 shows the comparison between the AAM-kriging emulator described in section 3 (Fig. 8(c)) and the two usual approaches (Fig. 8(a) and (b)). In addition, we tested a fourth method: the AAM-kriging emulator applied to a lower threshold exceedance than the actual guide-level value, to take a margin on the results obtained by the emulator (Fig. 8(d)). To achieve that, instead of creating an isoline at $\log(th)$ on our logarithmic dose, we created an isoline at $\log(th)/1.1$.

We compared the isolines of these different prediction methods to the one given by the Gaussian puff model for the 1000 simulations of our test sample. Four kinds of areas may be defined:

    – True-Positives: areas where both the predictor and the model forecast a threshold exceedance;

    – True-Negatives: areas where the predictor and the model do not forecast a threshold exceedance;

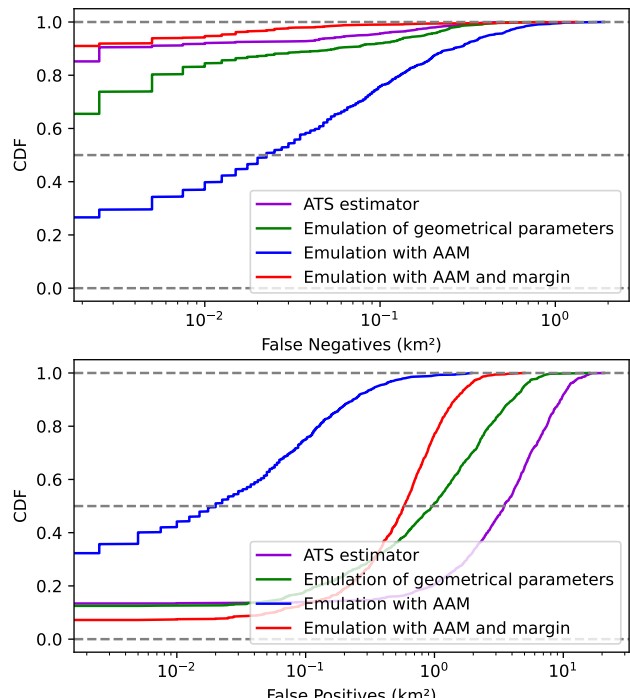

**Figure 9.** Cumulative distribution functions (CDFs) of the probability of obtaining false-positive and false-negative surfaces (in km$^2$) for the different predictors. These CDFs were estimated from a test sample of 1000 simulations. The upper plot corresponds to the distribution of false-negative surfaces, and the lower plot to the distribution of false-positive surfaces. For both plots, curves closer to the value 1 on the y-axis indicate better performance, as they correspond to a lower probability of significant false-positive or false-negative areas.

– False-Positives: areas where the predictor forecasts threshold exceedance, while the model forecasts the opposite;

– False-Negatives: areas where the predictor does not forecasts a threshold exceedance, while the model forecasts the
opposite;

These surfaces allow us to evaluate the performance of each method. Each output pair (model, predictor) is characterized by a certain amount of true-positive, true-negative, false-positive and false-negative areas.

Figure 9 presents the empirical cumulative distribution functions (CDFs) of the false-positive and false-negative surface areas (in km$^2$) over the 1000 test simulations. The x-axis represents the area, while the y-axis gives the proportion of simulations with
a surface area less than or equal to that value. For both curves, a faster rise towards 1 on the y-axis indicates better performance.

Table 3, in contrast, summarizes the total false-positive and false-negative areas accumulated across the 1000 simulations, for each prediction method. It provides an aggregate view of each method's overall behavior.

We observe that the ATS estimator, used as a very first response, produces the largest false-positive surface (10,665 km$^2$), with an extremely low false-negative area (0,030 km$^2$). This conservative behavior is expected, as the method prioritizes
avoiding any underestimation of risk.

| Estimator | Total surface ($km^2$) | |
| --- | --- | --- |
| | False-Positive | False-Negative |
| ATS estimator | 10.665 | 0,030 |
| Emulator of geometrical parameters | 3,846 | 0,056 |
| AAM-kriging emulation | 0,221 | 0,227 |
| AAM-kriging emulation with margin | 1,747 | 0,015 |

**Table 3.** Total surfaces (in $km^2$) of false-positive and false-negative areas over the 1000 test simulations, for each method used to predict the zones of dose threshold exceedance. The total area of the studied domain is 94.25 $km^2$. A false-positive surface corresponds to an area predicted as exceeding the threshold by the method but not by the reference model. A false-negative surface corresponds to an area where the method failed to predict a threshold exceedance that was actually present according to the model. Lower values in both categories indicate better predictive performance.

The emulator of geometrical parameters developed in a previous study reduces the false-positive surface (3,846 $km^2$) but increases false negatives (0,056 $km^2$), which are more critical from a public health perspective.

The AAM-Kriging emulator achieves a balanced result with low errors on both sides: 0,221 $km^2$ of false positives and 0,227 $km^2$ of false negatives. This method aims to match the reference isoline closely, without introducing conservative margins.

Finally, the margin-based AAM-Kriging method slightly increases the false positives (1,747 $km^2$) but drastically reduces the false negatives (0,015 $km^2$). This trade-off leads to a false-negative performance similar to the ATS estimator, while reducing false positives by more than a factor of eight.

## 5   Application

The combination of AAM and Kriging enables the construction of emulators capable of reproducing the model output in
approximately 0.005 seconds, compared to about 1 minute for the original pX dispersion model. This drastic reduction in computation time opens up new possibilities for operational use, especially in situations where rapid decision-making is crucial, such as nuclear emergencies.

One immediate application of the emulator is to facilitate probabilistic risk assessments. The need to include uncertainties in expert's assessments provided to decision makers in case of radiological emergencies was documented in the European
H2020 project CONFIDENCE (French, S. et al., 2020), and the use of ensemble simulations was recommended, provided the computational time was compatible with the operational time constraints of emergency situations (Bedwell, P. et al., 2020). Since a single prediction now requires only a few milliseconds, it becomes possible to generate thousands of simulations across the uncertainty space of the input parameters. This allows the uncertainty inherent in the atmospheric dispersion process to be fully propagated through to the final dose predictions. As an illustration, Figure 10 shows a map of the estimated probability
of exceeding a given dose threshold. Such probabilistic maps provide emergency managers with a richer understanding of the risk zones compared to a single deterministic prediction (Nagy, A. et al., 2020).

Beyond probabilistic analysis, we also developed a dedicated graphical interface (see Figure 11) to make the emulator easily accessible to non-expert users such as emergency responders. This tool allows users to quickly adjust key input parameters (such as wind speed, release height, or source term amplitude) and immediately visualize the resulting dose map. It thus supports both real-time response activities and training sessions, by offering an intuitive way to explore how changes in environmental or release conditions impact the dispersion of radioactive material.

Overall, the AAM-Kriging emulator greatly enhances both the reactivity and the robustness of operational decision support systems in the event of an atmospheric radiological release.

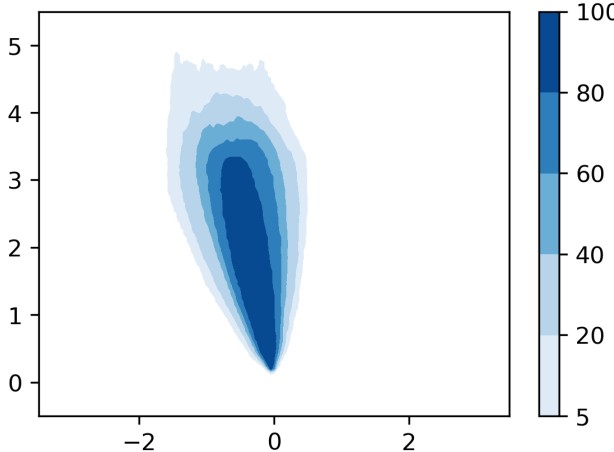

**Figure 10.** Example of a map which represent the estimation by the emulation with AAM of the probability of threshold exceedance after a nuclear accident.

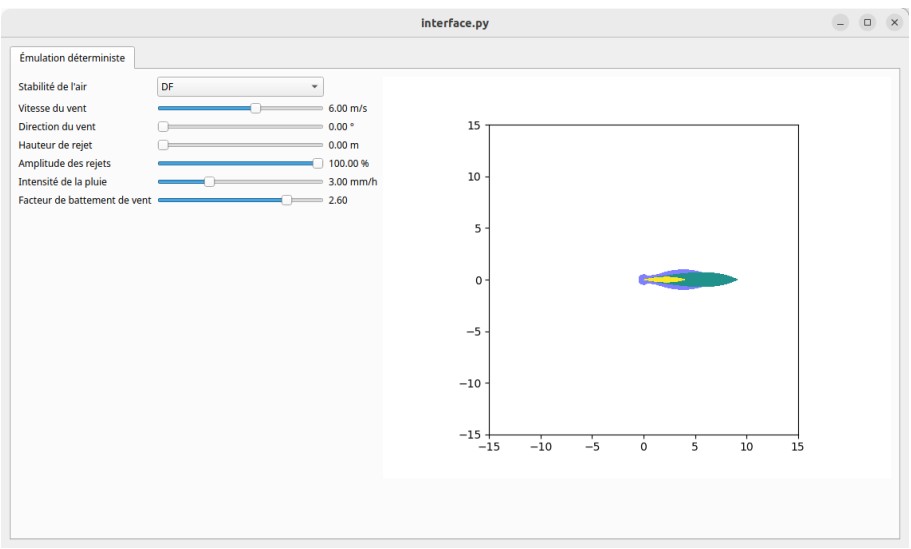

**Figure 11.** Example of a graphical interface created during the projet that enables users to directly observe how each input impacts the dose map.

## 6 Conclusion

AAM coupled with Kriging allows to create emulators which can reproduce the model output with a drastic reduction in computational time. The guide-level dose exceedance isoline obtained with the emulator is very close to the one obtained with the original model: the FMS between the two is under 0.8 in less than 17.6 % of our test sample. For an operational use, we recommend to take a margin by reducing the threshold exceedance of the dose. It will slightly increase the false-positives, but significantly decrease the false-negatives.

We obtained very similar results with different guide-levels and with the Doury dispersion model in weak diffusion.

This method of creating emulator is currently in an operationalization phase, to reproduce it on several other source terms, to create a catalog of emulator which cover scenarios among the ATS of ASNR.

This is the first time the AAM are used on a real and operational application. This study advances the state of the art in atmospheric dispersion by creating a new way to parameterize and predict quantity maps, which can be used in an operational 325 context with probabilistic approaches where hundreds of results must be obtained.

The main limitation of our approach is that in some rare cases, mainly when the wind module is low, the emulator's ability to reconstruct the model's predicted map is lower. However, these cases are also badly forecast by the physical dispersion model itself, and the error of the emulator would not necessarily exceed that of the pX model, were they compared to environmental observations. We think that these results would be improved by modifying the AAM construction method, which at some point 330 in its process uses a Euclidean distance to compare maps. However, some mathematical distances, such as the Wasserstein distance (Kolouri et al., 2017) for example, could be more suitable for comparing 2D dose maps within them. The AAM

method could then benefit from a modification to use other distances in its algorithm to go one step further in improving the application cases.

*Code and data availability.* The code used for generating and analyzing the simulation data is publicly available on Zenodo with the DOI: 10.5281/zenodo.14856799. The simulation data is hosted on Zenodo with the DOI: 10.5281/zenodo.14747261.

*Competing interests.* The authors declare that they have no conflict of interest.

*Author contributions.* R.P. developed the software, analyzed the data, contributed to the study design and methodology, and wrote the manuscript with input from all co-authors. S.G. and I.K. contributed to the study design and methodology. I.K. supervised the research.

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
