# Peer review of "Accurate and fast prediction of radioactive pollution by Kriging coupled with Auto-Associative Models"

_EGUsphere, 2024_

## Referee Comment (RC1)

**Review of: Accurate and fast prediction of radioactive pollution by Kriging coupled with Auto-Associative Models**

2025-04-04

The paper presents an innovative approach that combines Kriging and Auto-Associative encoders for fast estimation of the dispersion of radioactive pollutants in the atmosphere as support for the first response phase in a nuclear emergency.

The paper deserves publication once some minor revisions are addressed. In general, the paper requires a thorough review of the incorrect internal references of figures, paragraphs, etc., as well as an improvement of the English language.

Detailed notes:

- Section 1.1, first paragraph. The authors should present other possible approaches adopted from other entities to address these issues and not only reference their (and their institute's) previous paper on the subject.

- line 35. There is a missing reference for the pX code.

- line 55. double *both*

- Section 1.2, last paragraph. References to later sections are inconsistent, please check them. Section 5 is not mentioned or described.

- Table 1. The table requires additional explanations. The range of variation for each of the input variables is not discussed. For example, Release Height is taken between 0 and 100 m. Is it accurate to consider a release height of 0 m? Why cap it at 100 m? Explain the decisions taken and give references for the choices when possible.

- line 120. The description of the Kriging methodology is too vague. Improve it and add additional references for the reader to check.

- equation at line 126. $\forall$ symbols usage is inappropriate, rephrase as $\forall i \in [1, N]$ or similarly

- equation at line 126. $K(x, x_i)$ is not consistent with the $K$ at left-hand side, $x$ and $x_i$ should probably be switched. If this is not the case, please explain better what you are doing here.

- Section 3.2, last paragraph. Add some explanation and references for the BFGS algorithm and the procedure that you adopted.

- line 143. Please rephrase by stating the range of values.

- lines 145-148. The authors claim 9 coordinates is the sweet spot between accuracy and computational cost. A graph of the behavior of the accuracy (or, equivalently, of the error) as a function of the number of coordinates helps in conveying the claim.

- line 149. remove *Once these emulators built.*

- line 157. Missing closing parenthesis.

- lines 160-162. The figure reference is wrong. The description of the actual figure 4 is missing. The last phrase should be rewritten with additional explanation.

- Figure 7 caption. The numbers seem to be coming from figure 4, and they do not correspond to the description in line 184. Please check which ones are correct.

- line 188-189. Cases 3 and 4 seem to be switched with respect to the description above.

- line 221. Please improve the description of figure 9 and table 3, they are quite complex to understand and deserve a little bit more description.

- line 240. Replace *used as a training purpose* with *used for training purposes.*

- Section 5. The description of the section is very succinct. Please extend and illustrate better the objectives and results.

- line 243. Replace *allow* with *allows.*

---

## Referee Comment (RC2)

**Review of "Accurate and fast prediction of radioactive pollution by Kriging coupled with Auto-Associative Models" (egusphere-2024-3838)**

In this manuscript, the authors propose an approach to predict the spatial distribution of radioactive pollution using a novel emulation methodology applies to numerical model outputs. The emulation methodology involves constructing a reduced-dimension representation of the high dimensional model outputs using a method called auto-associative models (AAMs). The mapping between the numerical model parameters and the reduced-dimension representation is constructed using a kriging approach. The authors compare their method to existing approaches for predicting nucleotide exposure in a French context and find it to be both accurate (in terms of false-negative and false-positive rates) and fast to run.

The significance of the work is well justified, and the emulation approach is well principled. I also find the application to be principled and, on the whole, well executed. However, I have major comments regarding the clarity and detail of the exposition, given below. I also think the authors need to do more to justify the use of AAM in place of the more traditional approach of using principal component analysis (PCA). Finally, I have some minor typographical comments.

My major comments are as follows:

1. Regarding the use of AAM:

   (a) The authors provide almost no technical detail on AAM at all. I think it is impossible to understand the results without at least some technical detail.

   (b) The authors claim that AAM is better able to "capture nonlinear structures" than PCA. What does this mean? Non-linear in what? PCA involves a linear combination of basis vectors, but each basis vector can contain non-linear structures in the original parameter space.

   (c) The authors ought to provide some evidence that AAM is better than PCA for their specific problem. A very simple way would be to repeat the validation analysis in Section 4.1.1 using PCA and compare the metrics. I think that would suffice, avoiding the authors the need to reconstruct the entire emulator using PCA.

2. Regarding the construction of the emulator:

   (a) Can the authors expand on how the dimension of the AAM was chosen? For example, can the metrics in Section 4.1.1 be given for different dimensions? Such a discussion would be valuable for someone wanting to use the method for a different problem.

   (b) Are the emulators fit independently to each AAM parameter? This can be justified for PCA through the orthogonality of the principal components, is there a similar justification for AAM?

3. Regarding the validation metrics:

   (a) I think that in Figure 7, the histogram, this is the FMS across the training samples, is that right? It would be good to clarify this in the text.

   (b) I find the FMS metric in Section 4.1.1 to be hard to interpret. I much prefer the separate treatments of false positives and false negatives that eventually occurs for the emulator in Section 4.2. Can the authors just use this here?

I think this is important because it speaks to how conservative the method is in different ways. Indeed, I think the whole application could be better structured by discussing how to manage the trade-off between false positives and false negatives (where I note that in the end the author's method was best for both!)

(c) In Section 4.1.2, the SMSE appears to equal to the conventional $1 - R^2$ from linear regression. Is this correct? If so, why is the SMSE for Score 9 so bad, when the predictions look okay in Figure 5?

(d) In Section 4.2, I don't understand the x-axis for Figure 9 and its relationship to Table 3. Can the authors explain the metrics in more detail?

My minor comments are as follows:

1. The abstract states "The main limitation of emulation methods is that they can only predict scalar quantities." This may be true for existing methods for radioactive pollution, but it is not true in general: many moderate- and high-dimensional emulators have been constructed for vector-valued outputs. The authors even cite some papers in atmospheric science in their introduction to this extent.

   A similar statement occurs around line 53, when in the very following paragraph the authors give some examples of vector-valued outputs.

2. The authors may wish to consider whether to cite the paper by Cartwright et al. (2023) which uses a related approach to emulating vector-valued outputs through neural networks.

3. It may be worth citing a standard text of kriging in Section 3.2, e.g., Cressie (1993), for some of the mathematical details.

4. In Figure 6, the legend is extremely small.

Some typographical comments:

1. When giving multiple parenthetic citations, please format these as (Name et al., 2024; Name et al., 2025), rather than (Name et al., 2024), (Name et al., 2025).

2. Line 35: there is a stray () here.

3. Line 62: "sets of map" should be "sets of maps"?

4. Table 1: the formatting of the units is inconsistent.

**References**

Cartwright, L., Zammit-Mangion, A., and Deutscher, N. M. (2023). Emulation of greenhouse-gas sensitivities using variational autoencoders. *Environmetrics*, 34(2):e2754.

Cressie, N. (1993). *Statistics for Spatial Data*. John Wiley & Sons.

---

## Author Response (AR1)

**Author Response to Reviewer #1**

We thank Reviewer #1 for their constructive comments. Please find below our responses to each point raised.

**General comments:**

We thoroughly reviewed and corrected incorrect internal references (figures and sections), and improved the English language throughout the paper.

**1. Section 1.1 − Broaden context beyond internal work**

We added a citation to the IAEA report "Considerations in the Development of a Protection Strategy for a Nuclear or Radiological Emergency" (2021) to broaden the contextual background beyond our own institute's previous studies.

**2. Line 35 − Missing reference for pX code**

We added two references for the pX code in Section 1.1 to clarify its origin and documentation.

**3. Line 55 − Typo: double "both"**

Corrected.

**4. Section 1.2 − Incorrect section references**

We fixed all references to later sections and ensured that the reference to Section 5 is correct and relevant.

**5. Table 1 − Clarify parameter ranges**

We added explanations in the text and in the table caption for each parameter range. For example, we clarified that 0–100 m for release height is consistent with typical nuclear accident scenarios and excludes extreme or implausible configurations. We also provided references to justify these ranges.

**6. Line 120 − Improve Kriging description**

We revised Section 3.2 to improve the explanation of Kriging. While we chose not to enter into too much technical detail, we added references to Rasmussen and Williams (2006), Roustant et al. (2012), and Girard et al. (2016) for readers seeking more information.

**7. Equation line 126 − Improve mathematical notation**

We corrected the notation to $\forall i \in \{1, \ldots, N\}$ and ensured consistency between left- and right-hand sides. The explanation now matches the equation.

**8. Section 3.2 − Add BFGS reference and explanation**

We ultimately decided not to mention BFGS to avoid unnecessary detail at that point in the manuscript.

**9. Line 143 − Specify range values**

We added a sentence in Section 3.3 to explain that dose values vary by up to a factor of $10^5$ depending on distance.

**10. Line 145–148 – Justify AAM dimension choice**

We added Figure (figure 1) and an accompanying paragraph in Section 3.3 to explain how the number of AAM coordinates was selected.

**11. Line 149 – Typo**

Removed "Once these emulators built."

**12. Line 157 – Missing closing parenthesis**

Corrected.

**13. Lines 160–162 – Figure reference and description**

We clarified the reference to Figure 4 and expanded its caption.

**14. Figure 7 caption – Inconsistencies with text**

We corrected the figure (previously incorrect due to file mix-up) and aligned the caption and text.

**15. Line 188–189 – Case 3 and 4 descriptions swapped**

Corrected.

**16. Line 221 – Improve Figure 9 and Table 3 explanations**

We expanded the captions and added explanatory paragraphs at the end of Section 4.2 to help interpret both elements.

**17. Line 240 – Typo**

"allow" was replaced with "allows".

**18. Section 5 – Too succinct**

We significantly expanded Section 5 to better explain the objectives and results. We illustrated practical applications and cited the CONFIDENCE project to emphasize the relevance for operational use.

**19. Line 243 – Typo**

Corrected.
* * *
**Author Response to Reviewer #2**

We thank Reviewer #2 for the constructive and detailed feedback. Please find below our point-by-point responses.

**Major Comments**

**1. Regarding the use of AAM:**

(a) More context on the method has been introduced in Section 1.2. However, we intentionally kept Section 3.1 concise to avoid going too deep into technical details. We provide relevant references for readers who wish to explore further.

(b) Section 1.2 was significantly expanded to clarify what we mean by "non-linear structures". In particular, we now explicitly contrast the linear subspace assumption of PCA with the curved manifold approximation used in AAM.

(c) Section 1.2 also includes a reference to Girard et al. (2020), which compares AAM and PCA for reconstructing radiological dispersion maps — a case very similar to our own.

**2. Regarding the construction of the emulator:**

(a) We now include Figure~**??**, which shows the evolution of reconstruction error as a function of AAM dimension. The discussion is expanded in Section 3.3.

(b) Yes, the emulators are fit independently for each coordinate. This is now clarified in Section 3.3.

**3. Regarding the validation metrics:**

(a) We clarified in the text and caption that Figure 7 shows FMS results computed on the test set. We also corrected the figure, which was previously showing the wrong version due to a file mix-up. The caption of Figure 4 has also been improved for consistency.

(b) We did not modify the manuscript on this point, as the FMS is a well-established metric in the radioactive dispersion modeling community. A reference to its use in the CONFIDENCE project has been added to support this.

(c) The SMSE of some scores is strongly degraded due to a single outlier that is poorly reconstructed and falls outside the axes limits of Figure 5. To mitigate this, we added the 95% quantile of the relative absolute error in Table 2, providing a more robust alternative metric that is less sensitive to extreme values.

(d) To clarify Figure 9, we expanded its caption and improved the caption of Table 3. Additional explanation was added at the end of Section 4.2 to help interpret both the figure and the table.

**Minor Comments**

1. We modified the abstract and Section 2 to clarify that scalar-only emulators are common in radioactive dispersion but not a limitation of emulators in general. The sentence around line 53 was removed for consistency.

2. We chose not to cite Cartwright et al. (2023) to keep the focus on directly relevant literature and avoid overloading the manuscript with peripheral references.

3. We chose not to cite the other suggested reference, as we have not read it in detail and preferred to reference sources that were directly used and understood in our work.

4. We expanded the caption of Figure 6 for clarity.

**Typographical Comments**

1. Parenthetic citations have been reformatted according to GMD style.
2. The stray parenthesis at line 35 has been removed.
3. "sets of map" corrected to "sets of maps".
4. Table 1 units have been harmonized.
* * *
**Response to Community Comments**

**Response to Juan Antonio Añel (CEC1):** Thank you for the reminder. The code DOI is now correctly included in the codedataavailability section of the revised manuscript.
* * *
We hope that the revised manuscript addresses all concerns raised. We thank the reviewers and the community for their constructive comments.

---

## Author Response (AR2)

**Author Response to Editorial Comments**

We thank the Handling Topic Editor, Dr. Luke Western, for his careful reading of the revised manuscript and for the clear and constructive technical corrections. We have implemented all the requested revisions, as detailed below.

**1. Please write affiliation 2, ASNR, in full**

Done. Affiliation 2 is now written in full as *Autorité de Sûreté Nucléaire et de Radioprotection.*

**2. Throughout: Please ensure that citations contain only surnames, e.g., Leadbetter, S.J. et al., 2020 should be Leadbetter et al., 2020**

All in-text citations have been revised to remove first name initials. Citations now contain only surnames.

**3. Line 20: Please change LE et al., 2021 to Le et al., 2021**

Corrected. "LE" has been changed to "Le" in line 20.

**4. Throughout: Please ensure all units are in exponential format, i.e., m/s should be, e.g., m s^{-1}. Please also remove the points between units in Table 1, i.e., m s^{-1} instead of m.s^{-1}.**

All units have been revised to exponential format throughout the text and tables. Points between units have been replaced by proper spacing, following the journal style.

**5. Table 2: Please write as e.g., 1×10^{-1} rather than 1e-1 (remove scientific format)**

Scientific notation has been replaced by exponential notation in Table 2, as requested.

**6. Figure 9: please ensure that the legend does not overlap in the lower panel. As the legend is the same for both panels, a single legend in one figure should suffice.**

The legend in the bottow graph has been removed to avoid any overlap and ensure clarity.

**7. For all figures: Please ensure that figures with multiple panels are labelled as a), b), etc.**

All figures with multiple panels now include panel labels ((a), (b), etc.).

**8. Figure 8: It's not clear what the different colours mean here. Please explain in the figure caption.**

The caption of Figure 8 has been expanded to explain the meaning of each color, and to indicate that the color scheme is consistent with that used in other figures for clarity and comparison.

**9. Table 3 and lines 280–295: There are a mixture of decimal points and commas here. I assume that these should be decimal points throughout?**

Confirmed and corrected. All numerical values now consistently use decimal points.

**10. Code and data availability section: Please write the repositories as full citations and include in the bibliography, rather than only the doi.**

Full bibliographic entries for both the code and data repositories have been added to the `.bib` file and cited accordingly in the section. The DOIs have also been retained in the text to facilitate direct access.

**11. Line 341: Please write INTERNATIONAL ATOMIC ENERGY AGENCY in lower font**

Corrected. The agency name is now in sentence case.

**12. Line 372: Please change to "2nd edition"**

Corrected.

**13. Roustant et al. in the bibliography needs a doi**

A DOI has been added to the Roustant et al. reference.

**14. Swallow et al. in the bibliography needs a journal and doi**

Swallow et al. (2017) corresponds to an arXiv preprint. Since it has not been published in a peer-reviewed journal, we have retained it as a preprint and formatted the entry using the `@misc` type, including the arXiv identifier and URL.

**15. Tombette et al. in the bibliography needs a journal and doi**

This reference corresponds to a poster presented at the 4th European IRPA Congress and was not published in a journal nor assigned a DOI. We have reformatted it using the `@misc` type, indicating the presentation type and the conference as publisher.

We thank the reviewer again for their helpful and constructive feedback, which has contributed to improving the overall clarity and consistency of the manuscript.